



# Impact of terrestrial reference frame realizations on altimetry satellites orbit quality, global and regional sea level trends: case ITRF2014 versus ITRF2008

Sergei Rudenko[1,2], Saskia Esselborn[1], Tilo Schöne[1], and Denise Dettmering[2]

[1]Helmholtz Centre Potsdam - GFZ German Research Centre for Geosciences, Telegrafenberg, 14473 Potsdam, Germany
[2]Deutsches Geodätisches Forschungsinstitut der Technischen Universität München (DGFI-TUM), Arcisstr. 21, 80333 Munich, Germany

*Correspondence to:* S. Rudenko (sergei.rudenko@tum.de)

**Abstract.** A terrestrial reference frame (TRF) is a basis for precise orbit determination of Earth orbiting satellites, since it defines positions and velocities of stations tracking data of which is used to derive satellite positions. In this paper, we investigate the impact of the International Terrestrial Reference Frame realization ITRF2014, as compared to its predecessor ITRF2008, on the quality of orbits, namely, on root-mean-square (RMS) fits of observations and orbital arc overlaps of three altimetry

satellites (TOPEX/Poseidon, Jason-1 and Jason-2) at the time interval from August 1992 till April 2015 and on altimetry products computed using these orbits, such as single-satellite altimeter crossover differences, radial and geographically correlated mean sea surface height (SSH) errors, regional and global mean sea level trends. The satellite orbits are computed using Satellite Laser Ranging (SLR) and Doppler Orbitography and Radiopositioning Integrated by Satellite (DORIS) observations of a global network of stations.

We have found that using ITRF2014 generally improves orbit quality, as compared to using ITRF2008. Thus, the mean values of the RMS fits of SLR observations decreased (improved) by 2.4 and 8.8% for Jason-1 and Jason-2, respectively, but are almost not impacted for TOPEX/Poseidon, when using ITRF2014 instead of ITRF2008. The internal orbit consistency in the radial direction (as derived from arc overlaps) is reduced (improved) by 6.6%, 2.3%, and 5.9% for TOPEX/Poseidon, Jason-1, and Jason-2, respectively.

Single satellite altimetry crossover analysis indicates reduction (improvement) of the absolute mean crossover differences by 0.2 mm (8.1%) for TOPEX/Poseidon, 0.4 mm (17.7%) for Jason-1 and 0.6 mm (30.9%) for Jason-2 with ITRF2014 instead of ITRF2008. The major improvement of the mean values of the RMS of crossover differences (0.13 mm (0.3%)) has been found for Jason-2.

Multi-mission crossover analysis shows slight improvements in the standard deviations of radial errors: 0.1%, 0.2% and
1.6% for TOPEX/Poseidon, Jason-1 and Jason-2, respectively. The standard deviations of geographically correlated mean SSH errors improved by 1.1% for Jason-1 and 5.4% for Jason-2 and degraded by 1.3% for TOPEX/Poseidon.

The change from ITRF2008 to ITRF2014 orbits has only minor effects on the estimation of regional and global sea level trends over the 22 years time series from 1993 to 2015. However, on interannual time scales (3-8 years) large scale coherent trend patterns are observed that seem to be connected to drifts between the origins of the tracking stations networks. This leads





to uncertainties of interannual global mean sea level of up to 0.06 mm/year for TOPEX/Poseidon, 0.05 mm/year for Jason-1, and up to 0.12 mm/year for Jason-2. The respective changes of regional sea level trend on these time scales reach maximum values of 0.4 mm/year for TOPEX/Poseidon, of 0.5 mm/year for Jason-1 and of 1.0 mm/year for Jason-2.

## 1 Introduction

Precise information on positions and motion of points located on the Earth's surface is important for practical applications, such as positioning and navigation, and scientific investigations, such as Earth's rotation, plate tectonics, seismological deformations, precise orbit determination (POD) and some others. Precise positions and velocities of geodetic stations are provided by International Terrestrial Reference System (ITRS) realizations created by ITRS Combination Centres based on solutions provided by International DORIS Service (IDS), International GNSS Service (IGS), International Laser Ranging Service (ILRS) and
International VLBI Service for Geodesy and Astrometry (IVS). These solutions are derived from the analysis of Doppler Orbitography and Radiopositioning Integrated by Satellite (DORIS), Global Positioning System (GPS), Satellite Laser Ranging (SLR), and Very Long Baseline Interferometry (VLBI) observations. Three new ITRS realizations have been released recently. They are ITRF2014 (Altamimi et al., 2016), DTRF2014 (Seitz et al., 2016) and JTRF2014 (Abbondanza et al., 2017).

  A precise and stable terrestrial reference frame (TRF) is important for long-term consistency of altimetry measurements,
since it provides the basis for mapping sea level change to an accurate and stable coordinate system for calibration and validation and improved long-term monitoring of sea level changes (Fu and Haines, 2013; Wöppelmann and Marcos, 2016). The realization of a terrestrial reference system has been shown to have noticeable impact on the regional and global mean sea levels trends derived from altimetry (Beckley et al., 2007). While Beckley et al. studied differences related to the change from ITRF2000 to ITRF2005, here we analyse the impacts of the change from ITRF2008 to ITRF2014. Morel and Willis (2005)
showed that a 10-mm error in the $T_Z$ realization of a terrestrial reference frame can cause a systematic error of -1.2 mm in the derived mean sea level. Using TOPEX/Poseidon altimetry data, they estimated a precision of 3.0 mm in sea level and 0.37 mm/year in sea level trend in case of ITRF97. From the analysis of Jason-1 and Jason-2 orbits derived using GPS and SLR+DORIS observations in 2002–2014 in ITRF2008 instead of ITRF2005, Couhert et al. (2015) estimated a sea level trend error caused by these ITRF realizations of 0.05 mm/year (globally) and 0.3 mm/year (regionally) at the decadal time scale.
From their study, the sea level trend error caused by these ITRF realizations can reach 0.03 mm/year (globally) and 1 mm/year (regionally) at the interannual time scale. More recently, from the analysis of TOPEX/Poseidon observations for the period 1993 to 2004, Esselborn et al. (2018) have estimated a sea level trend error caused by ITRF2008 realization, as compared to ITRF2014 of 0.01 mm/year (globally) and 0.2 mm/year (regionally) at the decadal time scale and 0.03 mm/year (globally) and 0.2 mm/year (regionally) at the interannual time scale. Recently, Zelensky et al. (2018) investigated the impact of ITRF2014,
DTRF2014 and JTRF2014 on orbits of altimetry satellites TOPEX/Poseidon, Jason-1, Jason-2 and Jason-3 derived using SLR and DORIS measurements over 1993–2015. They found that replacement of ITRF2008 by ITRF2014 causes the impact on the regional sea level trend within ±0.3 mm/year.



The most widely used ITRF realizations are derived by the International Earth Rotation and Reference Systems Service (IERS) ITRS Product Center at Institut National de l'Information Géographique et Forestière, France. Therefore, in this paper we assess the impact of the new (ITRF2014) realization, as compared to its predecessor, ITRF2008 (Altamimi et al., 2011), on the orbits of three altimetry satellites, namely, TOPEX/Poseidon (from 23 September 1992 until 9 October 2005), Jason-1

(from 13 January 2002 until 5 July 2013) and Jason-2 (from 5 July 2008 until 6 April 2015), since these are the reference missions for sea level investigation (Quartly et al., 2017). We have computed orbits of these satellites (called GFZ VER13) using the ITRF2014 reference frame. We analyze these orbits, as compared to the GFZ VER11 orbits (Rudenko et al., 2017) of the same satellites derived within the second phase of the Sea Level project of the European Space Agency (ESA) Climate Change Initiative (CCI, Legeais et al. (2018)) using the ITRF2008 reference frame. For both sets of orbits, all other background

models for precise orbit determination and estimated parameters are the same. Both VER11 and VER13 orbits are derived using SLR and DORIS observations. We investigate the impact of ITRF2014, as compared to ITRF2008, on the root-mean-square (RMS) of residuals of SLR and DORIS observations used for orbit determination, the impact on the RMS and mean of single-satellite altimetry crossover differences, radial errors and on the global and regional mean sea level trends. On the contrary to the paper of Zelensky et al. (2018), we investigate the impact of the replacement of ITRF2008 by ITRF2014 also on two-day

arc overlaps in radial, cross-track and along-track directions and the geographically correlated mean sea surface height (SSH) errors. Moreover, we use 3.7- and 5-year longer time intervals for TOPEX/Poseidon and Jason-1, respectively, as those used by Zelensky et al. (2018). Additionally, we use another software and models for orbit computation and altimetry analysis. All this leads to slightly different results, as obtained by Zelensky et al. (2018).

The rest of the paper is organized as follows. A short description of the ITRF2014 and its differences with respect to

ITRF2008 is given in Sect. 2. A description of the models used for POD of altimetry satellites and the impact of the ITRF2014, as compared to ITRF2008, on the RMS fits of SLR and DORIS observations used for orbit determination as well as on the two-day arc overlaps in the radial, cross-track and along-track directions is discussed in Sect. 3. The impact of ITRF realizations on the RMS and mean of single-satellite altimetry crossover differences for the three satellites is presented in Sect. 4. The influence of the change from ITRF2008 to ITRF2014 on the radial orbit errors and geographically correlated mean sea surface

height errors as well as on the global and regional mean sea level trends is presented in Sect. 5 and 6, respectively. The main results of our study are summarized in Sect. 7.

## 2   ITRF realizations used for precise orbit determination

The detailed description of ITRF2008 and ITRF2014 is given in (Altamimi et al., 2011) and (Altamimi et al., 2016), respectively. The main differences of ITRF2014 with respect to ITRF2008 consist in:

- 6-year longer time span (2009.0 – 2015.0) used for the generation of the reference frame and, therefore, an increased number of stations and their occupations,

- using information from 36 new surveys performed since the release of ITRF2008; this resulted in employing 139 local ties for ITRF2014 instead of 104 for ITRF2008,



**Table 1.** The number of DOMES numbers and discontinuities and data span for DORIS and SLR stations in the ITRF2008 and ITRF2014.

| Observation type | ITRF2008 | | | ITRF2014 | | |
|---|---|---|---|---|---|---|
| | Number of DOMES numbers | Number of discontinuities | Data span | Number of DOMES numbers | Number of discontinuities | Data span |
| DORIS | 130 | 40 | 1993.0-2009.0 | 160 | 62 | 1993.0-2015.0 |
| SLR | 128 | 24 | 1983.0-2009.0 | 140 | 41 | 1983.0-2015.0 |

- enhanced modelling of non-linear station motions,

- exclusion of annual and semi-annual variations in station positions prior to the determination of station positions and velocities,

- post-seismic deformations are provided for stations affected by major earthquakes.

Since observations only from DORIS and SLR stations are used in our study, the following description concerns only these stations. Since an additional time span was used in ITRF2014 for SLR and DORIS stations, ITRF2014 contains 30 additional DOMES numbers for DORIS stations and 12 - for SLR stations, as compared to ITRF2008 (Table 1). ITRF2014 includes 22 additional discontinuities for DORIS stations and 17 - for SLR stations, as compared to ITRF2008. Moreover, ITRF2014 provides post-seismic deformation models for 13 DORIS and 10 SLR stations that have been used by us for this reference frame. No annual and semi-annual signals were applied by us neither for ITRF2008, nor for ITRF2014, since they are not a part of these realizations and were estimated for ITRF2014 by its authors internally for enhancing the velocity field estimation of the secular frame.

## 3   Impact of ITRF2008 and ITRF2014 realizations on the orbit quality

To perform our study, we have derived orbits of TOPEX/Poseidon (from 23 September 1992 until 9 October 2005), Jason-1 (from 13 January 2002 until 5 July 2013), and Jason-2 (from 5 July 2008 until 6 April 2015) at 12-day arcs with 2-day arc overlaps. Orbit computations were performed using "Earth Parameter and Orbit System - Orbit Computation (EPOS-OC)" software (Zhu et al., 2004) developed at GFZ. We use SLR and DORIS observations for all three satellites. To derive the satellite orbits, we use the same models, procedures and parameterization, as described in (Rudenko et al., 2017), but use two different ITRF realizations - ITRF2008 and ITRF2014. The main models used for orbit determination are given in Table 2. For the details on the models and procedures used for the POD the reader is referred to (Rudenko et al., 2014) and (Rudenko et al., 2017). The orbits of these satellites derived using ITRF2008 and ITRF2014 are called VER11 (version 11) and VER13 orbits, respectively.

The RMS fits of observations are an indication of accuracy of observations, models, reference frame realizations and parameterization used for POD. Since we use the same observations, models and parameterization to compute the VER11 and VER13 orbits and replace only ITRF realizations, the changes in RMS fits of observations indicate the impact of the change



**Table 2.** The main models used for orbit determination (for the details and references for the models, see Rudenko et al. (2014, 2017)

| Item | The model used |
| --- | --- |
| Earth gravity field model | EIGEN-6S4 (up to n=m=90) |
| Solid Earth and pole tides | IERS Conventions (2010) |
| Ocean tide model | EOT11a |
| Atmospheric tides | Biancale and Bode (2006) |
| Non-tidal atmospheric and oceanic gravity | GFZ AOD1B RL05 |
| Third bodies (Sun, Moon and 7 major planets) | DE-421 ephemerides |
| Radiation pressure model | CNES/IDS box/wing models |
| Earth radiation | Knocke model |
| Atmospheric density model | MSIS-86 |
| Polar motion and UT1 | IERS EOP 08 C04 (IAU2000A) with IERS daily and sub-daily corrections |
| Precession and nutation | IERS Convention (2010) |
| Tropospheric correction for DORIS data | Vienna Mapping Function 1 |

of ITRF realizations on the RMS fits of observations. From the analyses of the satellite orbits computed by us, we have not found notable impact of the ITRF2014, as compared to ITRF2008, on the RMS fits of SLR observations of TOPEX/Poseidon. Their mean value slightly increased from 1.96 to 1.97 cm, i.e. by 0.3%. However, the mean values of SLR RMS fits decreased (improved) from 1.19 to 1.16 cm, i.e. by about 2.4%, for Jason-1 and from 1.23 to 1.13 cm, i.e. by 8.1%, for Jason-2, when

using ITRF2014 instead of ITRF2008. The major reduction of the SLR RMS fits is obtained in 2009–2015 (Figs. 1-2), since ITRF2014 was derived using additional observations for this time span allowing more precise determination of station positions at this time span and station velocity over the whole time interval. In these figures, we use 52-week running mean in order to get rid of short-periodic variations in the RMS fits.

    The mean values of DORIS RMS fits are reduced (improved) for Jason-2 from 0.3490 mm/s to 0.3484 mm/s, i.e. by about

0.2%, when using ITRF2014 instead of ITRF2008. A more significant improvement of 0.3–1.0% is observed in 2012–2015. A rather small impact on DORIS RMS fits of Jason-1 is found. Thus, a small improvement (about 0.2%) of DORIS RMS fits is observed in 2010–2011 and a small degradation (about 0.3%) of DORIS RMS fits is observed in 2012–2013 for this satellite. The mean values of DORIS RMS fits are almost not changed for TOPEX/Poseidon, when using ITRF2014 instead of ITRF2008. However, an improvement of DORIS RMS fits of 1–3% is observed at about 20 arcs in 1993–1998, when using

ITRF2014 instead of ITRF2008. The number of accepted DORIS observations at these arcs is 1.2–2.5 times larger when using ITRF2014.

    The differences of satellite positions computed at arc overlaps serve as an indication of the internal consistency of satellite orbits. We have found from our analysis that the internal consistency of satellite orbits derived using ITRF2014 has improved, as compared to that one obtained using ITRF2008. Thus, the mean values of the radial arc overlaps decreased (improved) from

0.89 to 0.83 cm, i.e., by 6.6%, for TOPEX/Poseidon, from 0.79 to 0.77 cm, i.e., by about 2.3%, for Jason-1 and from 0.56





**Table 3.** Mean values of the RMS fits of SLR and DORIS measurements, two-day arc overlaps and the number of arcs used to compute these values for TOPEX/Poseidon (from 23 September 1992 until 9 October 2005), Jason-1 (from 13 January 2002 until 5 July 2013), and Jason-2 (from 5 July 2008 until 6 April 2015) VER11 and VER13 orbits derived at the time intervals specified. The percentage of the change from VER11 to VER13 orbits is given in parentheses (positive value indicates an improvement).

| Satellite and orbit version | SLR RMS fit (cm) | DORIS RMS fits (mm/s) | Radial overlap (cm) | Cross-track overlap (cm) | Along-track overlap (cm) | Number of arcs SLR/DORIS/overlap |
|---|---|---|---|---|---|---|
| TOPEX VER11 | 1.96 | 0.4778 | 0.89 | 6.49 | 3.48 | 494 / 459 / 433 |
| TOPEX VER13 | 1.97 (-0.3%) | 0.4776 (0.0%) | 0.83 (+6.6%) | 6.42 (+1.1%) | 2.71 (+22.0%) | 494 / 459 / 433 |
| Jason-1 VER11 | 1.19 | 0.3532 | 0.79 | 4.17 | 2.48 | 441 / 441 / 270 |
| Jason-1 VER13 | 1.16 (+2.4%) | 0.3532 (0.0%) | 0.77 (+2.3%) | 4.10 (+1.7%) | 2.29 (+7.9%) | 442 / 442 / 272 |
| Jason-2 VER11 | 1.23 | 0.3490 | 0.56 | 3.34 | 1.46 | 255 / 251 / 190 |
| Jason-2 VER13 | 1.13 (+8.1%) | 0.3484 (+0.2%) | 0.53 (+5.9%) | 3.23 (+3.4%) | 1.28 (+12.4%) | 254 / 251 / 190 |

to 0.53 cm, i.e., by 5.9%, for Jason-2, when using ITRF2014 instead of ITRF2008. The mean values of the cross-track arc overlaps are reduced (improved) from 6.49 to 6.42 cm, i.e. by 1.1%, for TOPEX/Poseidon, from 4.17 to 4.10 cm, i.e. by 1.7%, for Jason-1, and from 3.34 to 3.23 cm, i.e. 3.4% for Jason-2. The most significant reduction (improvement) is found for the along-track arc overlap, namely, from 3.48 to 2.71 cm, i.e. by 22.0%, for TOPEX/Poseidon, from 2.48 to 2.29 cm, i.e by 7.9%

for Jason-1 and from 1.46 to 1.28 cm, i.e. by 12.4% for Jason-2 (Table 3).

## 4   Impact of the change from ITRF2008 to ITRF2014 on the single satellite altimetry crossover differences

Single crossover analyses for all three missions have been performed based on ESA CCI Sea Level v2 ECV data (Legeais et al., 2018). The data is available with a 1-Hz sampling rate, and all corrections for instrumental and geophysical effects by the state-of-the-art models are provided with the data. For consistency reasons, we replaced some internal correction models, in

particular, the ocean tide and loading correction with the EOT11a tide model (Savcenko and Bosch, 2012) and the solid earth tides following the IERS 2003 Conventions (McCarthy and Petit, 2004). The altimeter crossover differences (ascending pass minus descending pass) are calculated in a 10-day step with GFZ's Altimeter Database and Processing System (ADS, Schöne et al. (2010)) for each test orbit (VER11 and VER13) separately. The global mean crossover difference and the associated RMS values are calculated after applying a 3-sigma test. In average, about 5000 valid crossover points are found with some annual

changes due to hemispheric change in sea ice coverage. For all three missions, the RMS of the crossover differences is around 5 cm. Notable also is the non-zero mean of the crossover differences which indicates a constant offset in sea level heights between ascending and descending tracks. Comparing VER11 and VER13 results the mean of the crossover differences becomes smaller for VER13, indicating smaller discrepancies between ascending and descending tracks. Also a slight improvement of



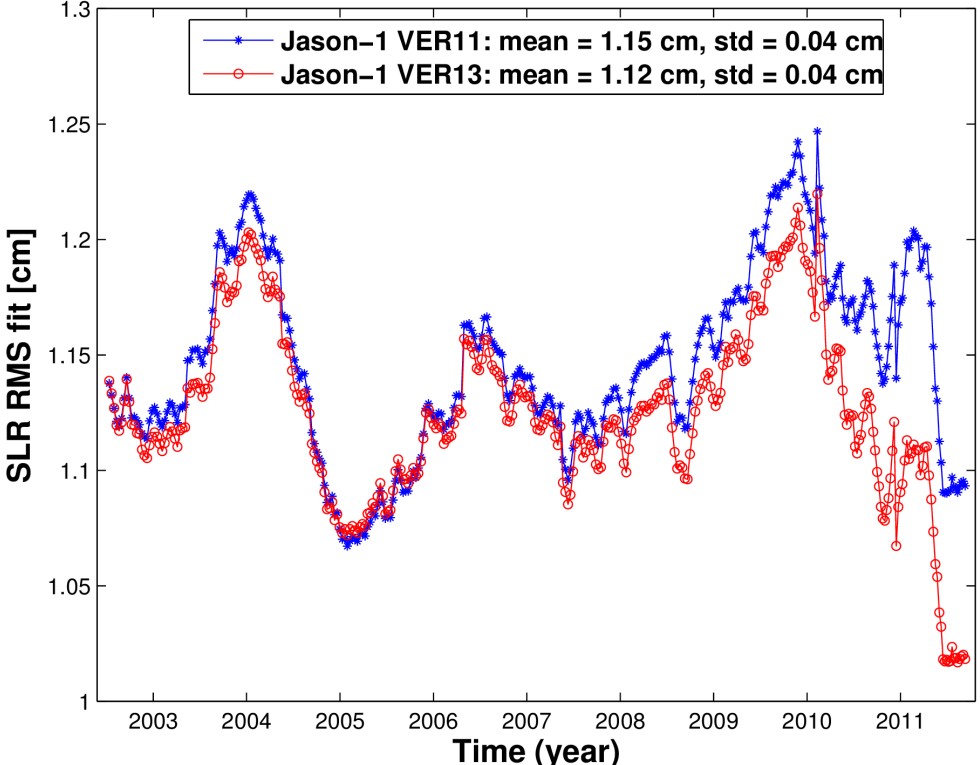

**Figure 1.** 52-week running mean of the RMS fits of Jason-1 SLR observations obtained using ITRF2014 (VER13 orbit) and ITRF2008 (VER11 orbit) from 13 January 2002 to 16 February 2012.

0.01–0.13 mm of the RMS of crossover differences is observed (Table 4) when replacing the VER11 orbit by the VER13 orbit. The quality of Jason-2 crossovers improves significantly, indicating a better performance of the ITRF2014-based VER13 orbit.

Also of interest are the differences of the mean crossover of VER11 minus VER13 over time (Fig. 3) differencing the crossover differences for the particular 10-day period. The differences for TOPEX/Poseidon are stable over the full mission period with only minor variations. They can be addressed to very small changes in the radial orbit component. For Jason-1 a small, negligible drift of 0.06 mm/year can be observed. This (positive) drift indicates that for the VER13 orbits the discrepancy between ascending and descending tracks becomes smaller with time. The drift in the difference for Jason-2 is much larger and reaches 0.31 mm/year. Notable also that starting around mid-2009 the differences for Jason-1 and Jason-2 show a sinusoidal signal with a period of about 120 days (the period is fixed to 10 days as per the analyses period) with amplitudes of around 4 mm. The larger scatter of crossover differences for the periods after middle of 2009 shown in Fig. 3 are due to the fact that station positions used for the computation of satellite positions are computed in case of using ITRF2008 by the extrapolation of station velocities derived by 2009.0 beyond this instant, while station velocities in case of using ITRF2014 are derived by



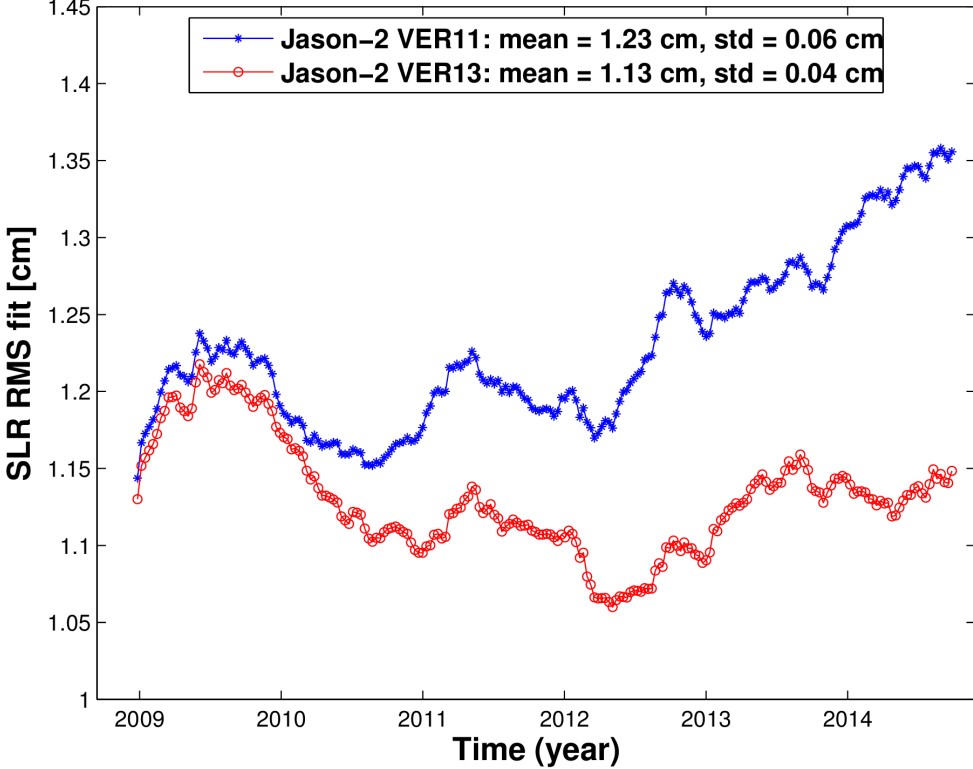

**Figure 2.** 52-week running mean of the RMS fits of Jason-2 SLR observations obtained using ITRF2014 (VER13 orbit) and ITRF2008 (VER11 orbit) from 5 July 2008 to 6 April 2015.

2015.0. This is an indication of an error, introduced by an older ITRF realization used beyond the time span at which it was derived.

## 5 Impact of the change from ITRF2014 to ITRF2008 on the radial orbit errors and geographically correlated mean sea surface height errors

5   In order to investigate the influence of using satellite orbits based on different realizations of the reference system on the precision and consistency of altimetry-derived sea level products, SSH crossover differences with a maximum time limit of two days are analyzed. For this purpose, a global multi-mission crossover analysis (MMXO) as described by Bosch et al. (2014) is applied to derive radial errors as well as geographically correlated error patterns for all three missions and for both orbit solutions. The comparison of the results obtained using ITRF2008 and ITRF2014 reveals valuable information on the product





**Table 4.** Statistics of crossover differences for VER11 and VER13 orbits. For Jason-1 the geodetic phase is not considered due to the change of crossover point distribution. The values are means over all 10-day analyses in [mm].

| Satellite orbit | Reference system | Mean of crossover differences | RMS of crossover differences | Number of 10-day cycles analysed |
|---|---|---|---|---|
| TOPEX/Poseidon VER11 | ITRF2008 | -2.73 | 50.13 | 454 |
| TOPEX/Poseidon VER13 | ITRF2014 | -2.51 | 50.08 | 454 |
| Jason-1 VER11 | ITRF2008 | 2.26 | 50.29 | 368 |
| Jason-1 VER13 | ITRF2014 | 1.86 | 50.28 | 368 |
| Jason-2 VER11 | ITRF2008 | 1.88 | 49.39 | 245 |
| Jason-2 VER13 | ITRF2014 | 1.30 | 49.26 | 245 |

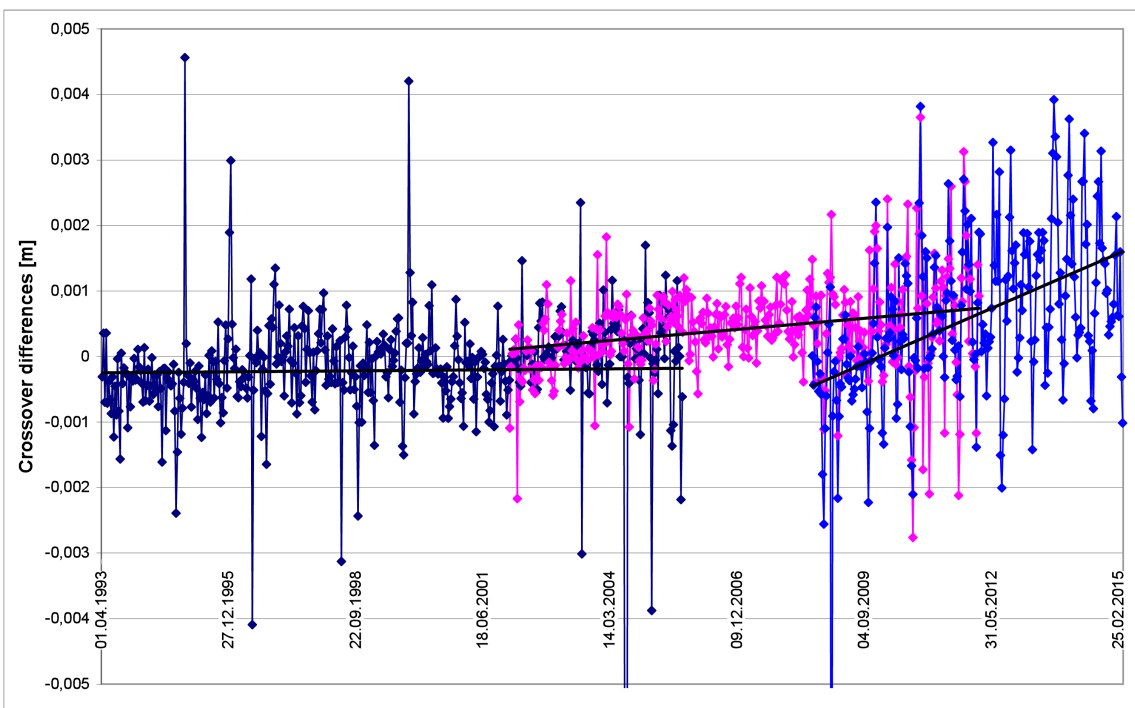

**Figure 3.** Difference of the crossover differences for all three missions, between VER11 and VER13. Values are in [m]. The X-axis shows time from 1 April 1993 until 25 February 2015.

quality and consistency for different periods. Similar investigations are performed by Rudenko et al. (2014) and Rudenko et al. (2016) for studying the impact of geopotential models and ocean and atmospheric de-aliasing products, respectively.



**Table 5.** Standard deviations of radial errors obtained using VER11 orbits based on ITRF2008 and VER13 orbits based on ITRF20014 of three satellites and their differences (positive values indicate improvements for orbits computed using ITRF2014)

| Satellite | VER11 ITRF2008 (cm) | VER13 ITRF2014 (cm) | Difference VER11-VER13 (cm) | Difference VER11-VER13 (%) |
|---|---|---|---|---|
| TOPEX | 1.486 | 1.485 | 0.001 | 0.1 |
| Jason-1 | 1.567 | 1.564 | 0.003 | 0.2 |
| Jason-2 | 1.103 | 1.086 | 0.017 | 1.6 |

For all three missions, slight improvements in the standard deviations of radial errors are obtained through the usage of ITRF2014 orbits as can be seen in Table 5. The choice of the reference system has only a small impact on the overall scatter of the radial errors and changes the standard deviations by less than one millimeter for all missions. However, whereas for TOPEX and Jason-1 the improvement is less than 1%, for Jason-2 an improvement of 1.6% is visible. This larger relative improvement

5  is partly related to the smaller scatter of radial errors. However, it is expected that the different behavior is also related to the measurement period of the missions. Thus, in order to access the temporal evolution of these values, standard deviations for each calendar year are computed. These values are plotted in Fig. 4 and reveal clear trends for Jason-1 and Jason-2. After 2010, significant improvements for all missions are visible reaching a maximum of nearly 3% for Jason-2 in 2014.

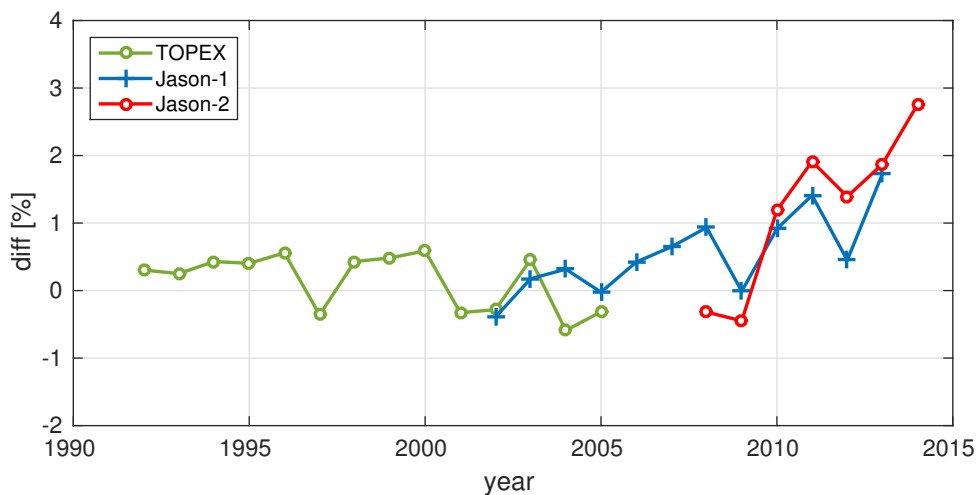

**Figure 4.** Relative difference (VER11-VER13) in the standard deviation of radial errors per year for three missions: TOPEX (green), Jason-1 (blue), and Jason-2 (red). Positive values indicate improvements for orbits based on ITRF2014.

For many sea level applications, the most harmful errors are those with a fixed geographical pattern. The MMXO provides
10  geographically correlated mean SSH errors (GCE) for all missions involved (Dettmering and Bosch, 2010; Dettmering et al.,



**Table 6.** Standard deviations of geographically correlated mean SSH errors obtained using VER11 orbits based on ITRF2008 and VER13 orbits based on ITRF20014 of three satellites and their differences (positive values indicate improvements obtained for the orbits computed using ITRF2014)

| Satellite | VER11 ITRF2008 (mm) | VER13 ITRF2014 (mm) | Difference VER11-VER13 (mm) | Difference VER11-VER13 (%) |
|---|---|---|---|---|
| TOPEX | 2.161 | 2.190 | -0.029 | -1.3 |
| Jason-1 | 2.188 | 2.165 | 0.023 | 1.1 |
| Jason-2 | 1.519 | 1.437 | 0.082 | 5.4 |

2015). The change from ITRF2008 to ITRF2014 for orbit computation also influences the GCE. Figure 5 displays the GCE for VER13 orbits (based on ITRF2014) as well as the GCE differences to those of VER11 orbit solutions. One can see that for all three missions, the GCE remain below about 1 cm and the change in ITRF accounts for less than 2 mm differences (positive as well as negative). Over the entire globe, the improvement yields 1.1% for Jason-1 and 5.4% for Jason-2, and a degradation
of 1.3% for TOPEX (in terms of reduction in standard deviation as it can be seen from Table 6).

## 6  Impact of the change from ITRF2008 to ITRF2014 on regional and global mean sea level

We investigate the interannual signals and long-term trends of the regional and global mean sea level from altimetry related to the change from the ITRF2008 to ITRF2014. Since the radial orbit component maps directly in the sea level measurement it is possible to study the effect of the improved ITRF on global and regional sea level from altimetry by analysing orbit data only.
In order to avoid the effects of noise and of the orbit parameterization, this study focuses on time scales of more than one year.

We evaluate the VER11 minus VER13 radial orbit differences sampled over the oceans. The orbits calculated in ITRF2014 are converted to the ITRF2008 system by a Helmert transformation by applying the transformation parameters from Altamimi et al. (2016). Jason-1 data from the geodetic phase (starting in May 2012) are excluded from the analysis. The orbit differences are mapped cycle-by-cycle along-track and are interpolated on a 1°x1° grid. From these data a global mean time series over the
ocean is inferred. From the global mean and the 1°x1° mapped time series, RMS and trend values are estimated, as described by Esselborn et al. (2018).

A measure of the amount by which the radial components of the two orbits are differing is the RMS value per cycle (Fig. 6). Its mean value is 1.8 mm for the combined TOPEX, Jason-1, and Jason-2 VER11 minus VER13 series which corresponds to 3% of the RMS value per cycle of the corresponding sea level data from altimetry (Table 7). While the values for most of
the TOPEX time series and also for the first few years of the Jason-1 series are below the mean, they are increased for the interleaved orbits of the TOPEX series (after mid of 2002) and after mid of 2006 for the Jason-1 and also for the Jason-2 series.





**Figure 5.** Geographically correlated mean SSH errors for three missions based on ITRF2014 orbits (left column) and VER11-VER13 differences (right column) for TOPEX (top), Jason-1 (middle), and Jason-2 (bottom).

The impact of the change of the ITRF solution on the estimated global mean sea level is minor. The RMS of the global mean radial differences over the ocean is 0.3 mm, which corresponds to 2% of the RMS of the global mean sea level from altimetry over the corresponding period (Table 7). The RMS of the global mean radial orbit differences over the ocean is slightly higher for the Jason-2 than for the TOPEX and Jason-1 missions. The global mean radial orbit differences are of the order of 0.5 mm for TOPEX and of 0.3 mm for Jason-1 and Jason-2. This offset between the two ITRFs is consistent with a slight shift (a few




**Table 7.** RMS per cycle, RMS and trend of the global mean over the ocean and maximum regional (absolute) trend values from VER11 minus VER13 radial orbit differences for the combined TOPEX, Jason-1 and Jason-2 series and for sub-series. The percentage of the given value relative to the values derived from altimetry data is given in brackets for comparison.

| Mission, time span | RMS per cycle [mm] | RMS of global mean difference [mm] | Trend of global mean difference [mm/y] | Regional trend up to [mm/y] |
|---|---|---|---|---|
| TOPEX/Jason-1/Jason-2 | 1.83 (3%) | 0.33 (2%) | -0.00±0.00 (0%) | 0.14 |
| TOPEX I (April 1993 to May 1997) | 1.82 (3%) | 0.19 (3%) | -0.06±0.01 (2%) | 0.38 |
| TOPEX II (June 1997 to September 2005) | 1.89 (3%) | 0.23 (3%) | 0.04±0.01 (1%) | 0.31 |
| Jason-1 I (January 2002 to October 2007) | 1.74 (3%) | 0.25 (4%) | 0.05±0.01 (2%) | 0.34 |
| Jason-1 II (October 2007 to February 2012) | 2.04 (3%) | 0.38 (6%) | -0.05±0.02 (2%) | 0.50 |
| Jason-2 I (July 2008 to March 2012) | 2.09 (4%) | 0.49 (9%) | -0.02±0.04 (1%) | 1.01 |
| Jason-2 II (March 2012 to April 2015) | 1.97 (4%) | 0.57 (11%) | 0.12±0.06 (4%) | 0.81 |

mm) of the VER13 origin from the South Pacific in the direction of Eurasia relative to the origin of the VER11 orbits. Such a shift is most probably related to slight changes of the positions of the tracking stations network.

The spectral analysis of the global mean radial orbit differences over the ocean shows that most of the energy can be found for periods of less than 110 days, however, this analysis is focusing on the interannual to decadal time scales. The low-pass
filtered time series of the global mean VER11 minus VER13 radial orbit differences over the ocean is shown in Fig. 7. The global mean sea level trend over the oceans is not affected by the use of the ITRF2014 realization for POD (Table 7). However, the global mean of the VER11 minus VER13 series (Fig. 7) exhibits interannual to decadal scale variability. From the inflection points of the time series, we identify the following four periods which are covered by the denoted missions:

1. April 1993 to May 1997 (TOPEX I),
2. June 1997 to October 2007 (TOPEX II, Jason-1 I),
3. October 2007 to March 2012 (Jason-1 II, Jason-2 I),
4. March 2012 to April 2015 (Jason-2 II).

For these periods, the trends of the global mean radial orbit differences range between -0.06 mm/year and 0.12 mm/year which corresponds to up to 4% of the sea level trend from altimetry over the corresponding periods (Table 7). The geographical
patterns of the trends for these four periods are given in Fig. 8. Note, that the trend patterns for Jason-1 I and Jason-2 I (not shown) resemble closely the patterns for TOPEX II and Jason-1 II (shown). For the first two periods (up to 2007), the trend patterns are consistent with relative drifts of the Z-components of the origins with a change of the direction in 1997. The regional trends in this period reach maximum values of 0.3 to 0.4 mm/year. For the last two periods, relative drifts of the origin in the horizontal plane are dominating. The regional trends after 2007 reach maximum values of 0.5 to 1 mm/year.

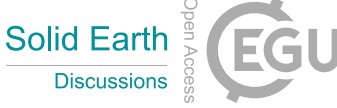



On the decadal scales, the error in the global mean sea level due to a TRF realization has decreased in the recent decades. Thus, Beckley et al. (2007) reported global mean sea level changes of 0.44 mm/year related to the change from ITRF2000 to ITRF2005. The change from ITRF2005 to ITRF2008 still lead to apparent global sea level drifts of 0.05 mm/year (Couhert et al., 2015) at the decadal time scale. The global mean sea level is hardly impacted by the change from ITRF2008 to ITRF2014

5  for the GFZ orbits. Based on GSFC orbits, Zelensky et al. (2018) estimated the error to be less than 0.04 mm/year. This is a major improvement with respect to the previous TRF realizations. Till 2008 apparent regional sea level drifts are mainly related to uncertainties of the z-coordinate of the origin. The pattern obtained by us is in line with the results from (Beckley et al., 2007), but the amplitudes became about four times smaller. After 2007 the uncertainties of the horizontal coordinates increasingly impact regional sea level drifts.

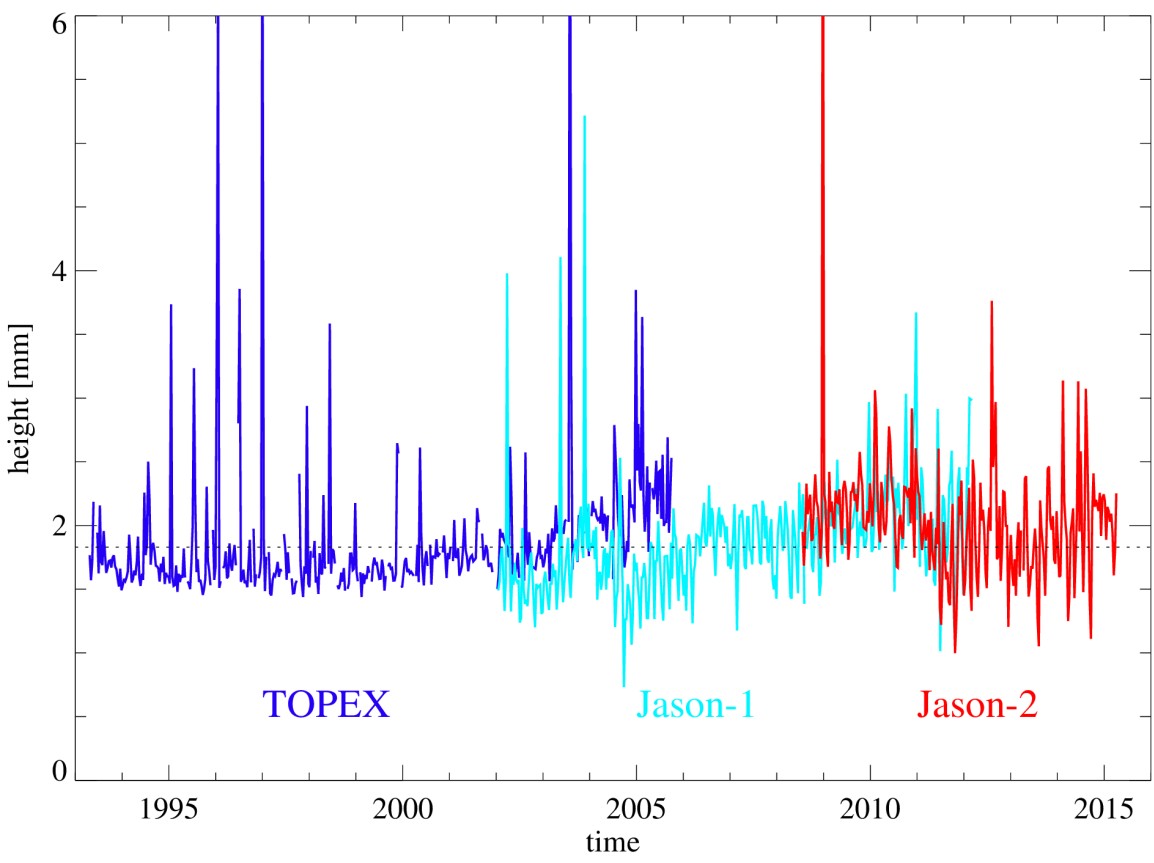

**Figure 6.** Global mean RMS per cycle of gridded radial orbit differences (VER11-VER13) for TOPEX (blue), Jason-1 (cyan) and Jason-2 (red). The mean value is marked by the dashed line.




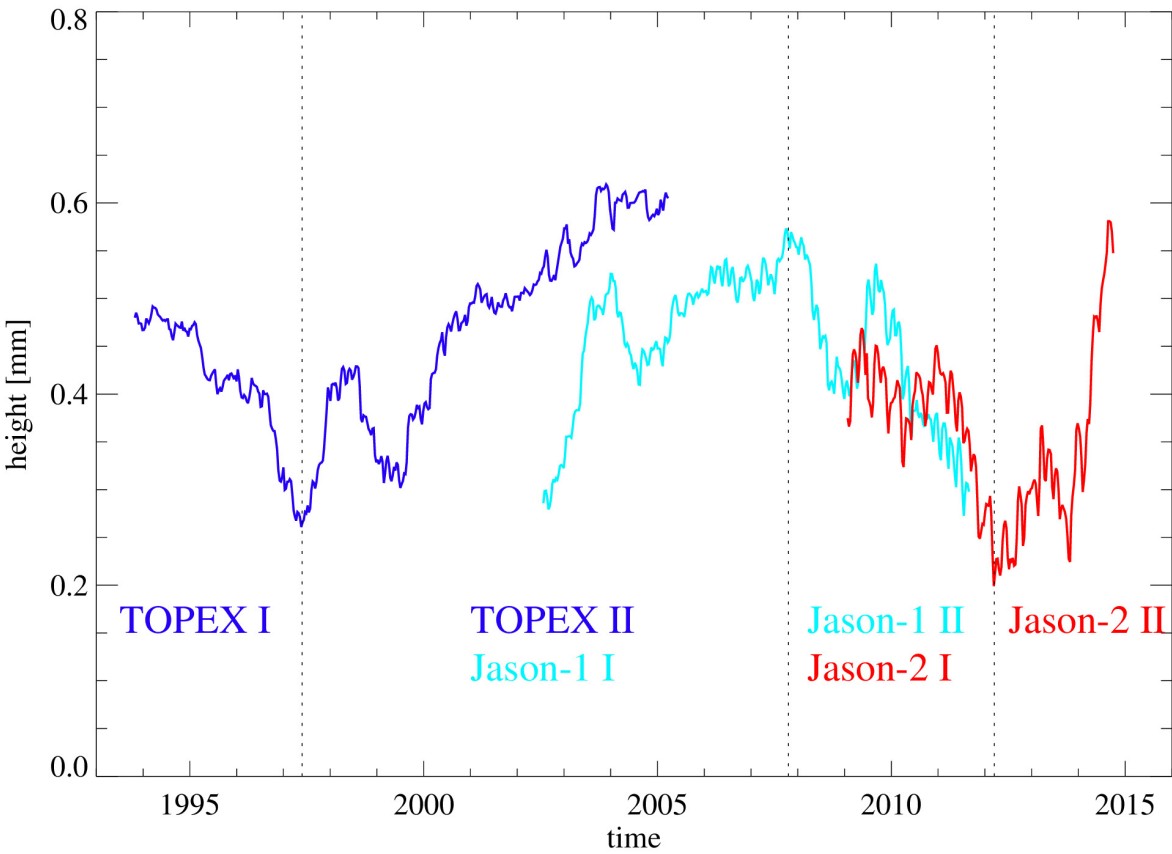

**Figure 7.** Mean radial height differences (VER11-VER13) over the global ocean low pass filtered by 1-year box-car filter for TOPEX (blue), Jason-1 (cyan) and Jason-2 (red). The sub-periods used for the calculation of trends are marked by dashed lines.

## 7    Conclusions

From the analysis of TOPEX/Poseidon (September 1992 to October 2005), Jason-1 (January 2002 to July 2013) and Jason-2 (July 2008 to April 2015) orbits computed by us using the ITRF2008 and ITRF2014 realizations, we have found that using ITRF2014 generally improves the orbit quality, as compared to using ITRF2008. Thus, the mean values of the RMS fits of SLR observations are reduced (improved) by 2.4% and 8.8% for Jason-1 and Jason-2, respectively, and are almost not impacted for TOPEX/Poseidon, when using ITRF2014 instead of ITRF2008. At the same time, the replacement of ITRF2008 by ITRF2014 has a minor impact (less than 0.1%) on the RMS fits of DORIS observations of TOPEX/Poseidon and Jason-1. A bit larger impact has been found for Jason-2, for which the mean values of DORIS RMS fits are reduced (improved) by about 0.2% over the whole time span (2008–2015) and a more significant improvement of 0.3–1.0% is observed in 2012–2015.





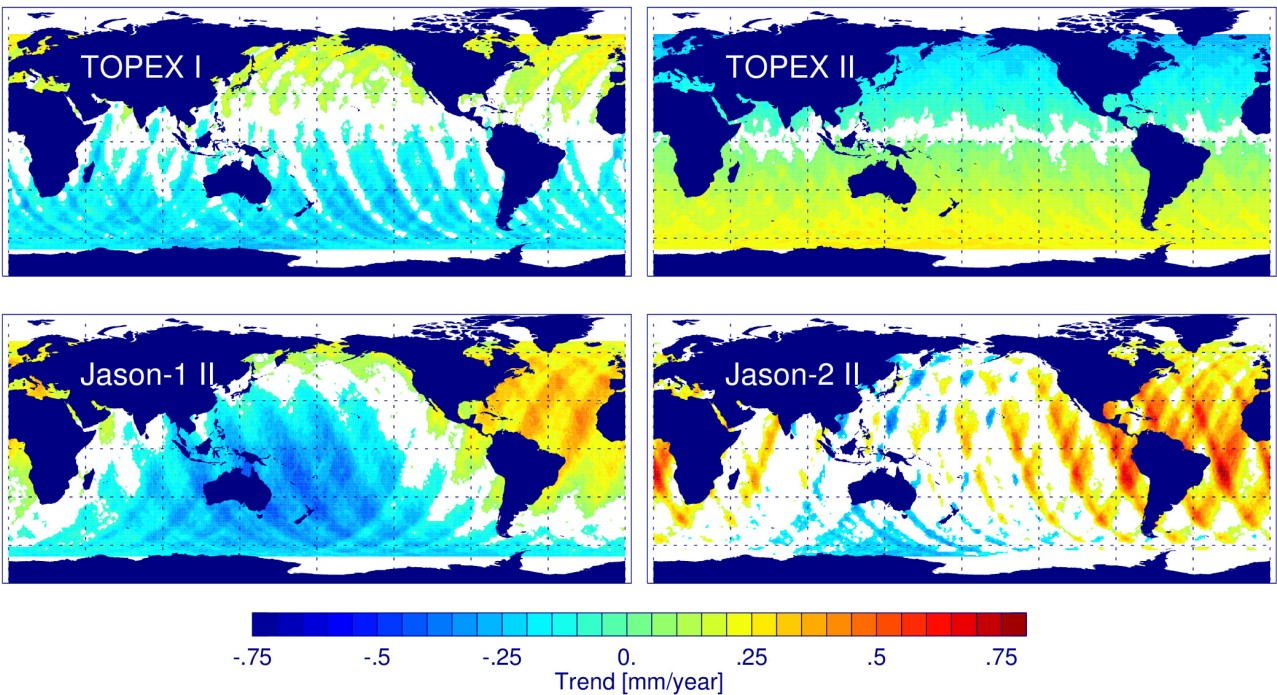

**Figure 8.** Trend differences of radial orbit components for VER11-VER13 for four periods. TOPEX I: Apr. 1993- May 1997, TOPEX II: June 1997-Sep. 2005, Jason-1 II: Oct. 2007-Feb. 2012, and Jason-2 II: Mar. 2012-Apr. 2015. Regions with formal errors larger than the fitted value are masked out (white). The global mean trend difference is given in Table 7.

The internal orbit consistency in the radial direction being important for altimetric applications and being characterized by the satellite position differences in this direction at two-day arc overlaps is reduced (improved) by 7.1%, 2.4% and 5.1% for TOPEX/Poseidon, Jason-1, Jason-2, respectively. The internal orbit consistency in the cross-track direction improved by 1.1% for TOPEX/Poseidon, 1.7% for Jason-1 and 3.4% for Jason-2 at the time spans analyzed, when using ITRF2014 instead of

5 ITRF2008. Even more significant improvement of the internal orbit consistency has been obtained in the along-track direction: by 22.0% for TOPEX/Poseidon, 7.9% for Jason-1 and 12.4% for Jason-2.

Single satellite altimetry crossover analysis indicates a reduction (improvement) of the absolute mean crossover differences by 0.2 mm (8.1%) for TOPEX/Poseidon, 0.4 mm (17.7%) for Jason-1 and 0.6 mm (30.9%) for Jason-2 with ITRF2014 instead of ITRF2008. The reduction of the mean of crossover differences indicates reduction of the discrepancies between ascending

10 and descending tracks, when using ITRF2014 instead of ITRF2008. The mean values of the RMS of crossover differences shows also reduction (improvement), when using ITRF2014 instead of ITRF2008, but at lower extend: by 0.05 mm (0.1%) for TOPEX/Poseidon, 0.01 mm (0.02%) for Jason-1 and 0.13 mm (0.3%) for Jason-2.

Multi-mission crossover analysis shows slight improvements in the standard deviations of radial errors through the usage of ITRF2014 for POD: 0.1% for TOPEX/Poseidon, 0.2% for Jason-1 and 1.6% for Jason-2. The standard deviations of ge-





ographically correlated mean SSH errors improved by 1.1% for Jason-1 and 5.4% for Jason-2, but degraded by 1.3% for TOPEX/Poseidon.

The change from ITRF2008 to ITRF2014 orbits has only minor effects on the estimation of regional and global sea level trends over the 22 years time series from 1993 to 2015. However, on interannual time scales (3-8 years) large scale coherent

trend patterns are observed that seem to be connected to drifts between the origins of the tracking stations networks. This leads to uncertainties of interannual global mean sea level of up to 0.06 mm/year for TOPEX/Poseidon, 0.05 mm/year for Jason-1, and up to 0.12 mm/year for Jason-2. The respective changes of regional sea level trend on these time scales reach maximum values of 0.4 mm/year for TOPEX/Poseidon, of 0.5 mm/year for Jason-1 and of 1.0 mm/year for Jason-2. The later value is at the edge of the user requirements for the error of the regional mean sea level (Ablain et al., 2015). While till 2008 regional

sea level drifts are mainly related to uncertainties of the z-coordinate of the origin, recently uncertainties of the horizontal coordinates have become increasingly important. The differences between the VER11 and VER13 orbits increase after 2009.0, since the ITRF2014 used for the computation of VER13 orbits provides station velocities more precisely than ITRF2008 used for the computation of VER11 orbits. This is due to the fact that ITRF2014 was derived at a time span until 2015.0, while ITRF2008 was computed at a time span only until 2009.0.

All analyses performed by us show that for the time spans from 2009.0 using a new ITRF realization (ITRF2014) improves the quality of orbits and altimetry products based on these orbits as compared to using the previous (ITRF2008) realization. This is in agreement with the results obtained by Rudenko et al. (2018) and Zelensky et al. (2018). Therefore, it is strongly recommended to use the new ITRS realization for precise orbit determination for the time span beyond the time instant of the end of the time interval used for the generation of a previous ITRS realization. This stresses also a need of periodical

reprocessing of altimetry satellite orbits using a new ITRS realization to get reliable and high accuracy altimetry products.

*Author contributions.* S.R. initiated this study, wrote Sect. 1-3, S.E. wrote Sect. 6, T.S. wrote Sect. 4, D.D. wrote Sect. 5. All authors contributed to Sect. 7

*Competing interests.* The authors declare that they have no conflict of interest.

*Acknowledgements.* This research was partly supported by the European Space Agency (ESA) within the Climate Change Initiative Sea Level

Phase 2 project and by the German Research Foundation (DFG) within the DGFI-project "Consistent dynamic satellite reference frames and terrestrial geodetic datum parameters" of the DFG Research Unit "Space-Time Reference Systems for Monitoring Global Change and for Precise Navigation in Space" and through grant CoRSEA as a part of the Special Priority Program (SPP) 1889 "Regional Sea Level Change and Society" (SeaLevel) and by the International Office of the BMBF under the grant 01DO17017 "Sea Level Change and its Hazardous Potential in the East China Sea and Adjacent Waters" (SEAHAP).



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
