# Peer review of "Impact of terrestrial reference frame realizations on altimetry satellites orbit quality, global and regional sea level trends: case ITRF2014 versus ITRF2008"

_Solid Earth, 2018_

## Referee Comment (RC1) · Anonymous Referee #1 · 13 Aug 2018

Review of "Impact of terrestrial reference frame realizations on altimetry satellites orbit quality, global and regional sea level trends: case ITRF2014 versus ITRF2008" by Sergei Rudenko, Saskia Esselborn, Tilo Schöne, and Denise Dettmering.

This paper discusses what the title says, namely the impact of a new reference system called ITRF2014 on sea level change observed by satellite altimetry. The conclusion is that the errors originating from orbit determination on the observed sea level reduce somewhat. The problem is investigated for TOPEX/Poseidon, and subsequent altimeters Jason-1 and Jason-2. In general, this paper is rather technical and unfortunately

not very exiting to read. I have some general comments to improve the quality.

Page 2, lines 1-2: I think it is better to not only mention the uncertainties, but also to mention the physical range of the sea level signal. Another aspect is that all used altimeter satellites only provide coverage between 66N and 66S, a part of the globe is missing, namely in the polar seas and also on the continental shelves.

Page 2, line 12-13: Merge the sentences

Page 2, line 17, "has been shown to have a noticeable impact" noticeable could be read as large or significant, but your study does not show this, the effect of orbit determination uncertainties on sea level rise observed by the altimeters is small. For semantic reasons you could better say, it is observable or detectable, but avoid to suggest that it is significant because it is not.

Page 2, line 18-19: "While Beckley et al . . . 2014" : In my opinion this sentence does not add new information to your paper. Remove it I would say.

Page 2, line 31-32: "They found that . . . mm/year" so this is more significant than what you mention, could you perhaps explain why their results differ from yours?

Page 3, line 2-3: "Therefore, in this paper . . . realization", this is repetition of information, these repetitive episodes make the paper uninteresting to read.

Page 3, line 7: Do your orbits for T/P, J-1 and J-2 satisfy the criteria of GDR-E which is the latest standard I believe? Where are the differences? Where are the common points? This should be discussed in your paper.

Page 4, line 1 under table 1: "non-linear station motions" please name it for what it is, it is called a (post) seismic deformation model in ITRF2014, is it the same as what you mention under the third point under table 1?

Page 5, line 2 under table 2: repair this sentence, for instance, the RMS of SLR to T/P did not change significantly when we switched from ITRF2008 to ITRF2014.

Page 5, line 15-16: did you focus on exactly the same ILRS stations and IDS beacons during the comparison? If this is not the case then I wonder about the realism of this comparison.

Page 5, line 17-18: It is not entirely clear what you mean with internal consistency of satellite orbits, this should be better explained. During orbit overlaps the trajectory is computed with the same gravity field, the same forcings apply on the satellite, the same station coordinates are used etc, so all possible common effect will cancel.

In table 3 I recommend to use ITRF 2008 and ITRF2014 consistently where you speak about VER11 and VER13. Below the table you describe exactly, in words, what the readers already see in the table. I do not understand why this information needs to be repeated.

Line 14 page 6, what makes a crossover point valid?

Lines 3-5 page 7: Did you ever investigate the choice of the time variable gravity model behavior on this? The same question could be asked about figure 2.

Figure 3 on page 9: this is too much of an excel like image in my opinion, the quality should be improved.

Table 4 and 5: either use ver11 and ver13, or use itrf2008 and itrf2014, but not the double referencing to ver11/itrf2008 and ver13/itrf2014 like you do now.

Line 2-3 below table 5: repetition of information, see earlier remarks.

Line 3 below table 5: TOPEX is the NASA Altimeter and TOPEX/Poseidon is the mission. I recommend that you use abbreviations like (T/P, J1 and J2). Please change all TOPEX references in the manuscript when you are referring to the mission. Only use TOPEX when you refer to the NASA altimeter on T/P.

Line 10 page 10: The concept of GCEs refers back to Rosborough 1987 who should be referenced. He was the first to start with geographically correlated orbit errors.

[Figure]

Page 11, line 14, use the \times symbol in LaTeX when you write 1x1

Line 9 and 10 on Page 13: TOPEX I and TOPEX II should in my opinion be referred to as TOPEX altimeter side A and side B.

Section 7, conclusions. I do not get the feeling that sea level rise estimates significantly change when ITRF2014 is introduced. The GMSL value is still 3.1+/-0.4 mm/yr (see the university of Colorado website) regardless whether we use the ITRF2008 or the ITRF2014 orbits. What you did improve are the DORIS and SLR residuals, but the improvements are small. Figure 8 shows that there are hemispherical like offsets developing over time, you may what to treat this as a geocenter type of discussion. This will affect the regional sea level estimates, which have larger ranges and error bars.

"Geographically" is hyphenated over a page break (16 and 17).

---

## Referee Comment (RC2) · E. C. Pavlis (Referee) · 12 Nov 2018

This article is presenting an important piece of work that even though it is not ground-breaking research, it is of great value to the users of altimetry data and products. When one reads this article one has the feeling that the intent of the authors is to record in high detail the work that was done to compare the difference in the "products" of the authors' lab between two reductions of the same (or nearly so) data with two different reference frame models. The analysis though is over different periods of time, something quite natural, since you only change reference frame models once the prior

version becomes obsolete. However, the presentation herein would have benefited from a slightly modified presentation of the differences: the one that is already in this version of the article, and a parallel one where the metrics are formed, presented and evaluated over exactly the same time period. This arrangement would remove the effect of the additional data used to develop the products beyond the validity period of the old TRF, along with any issues that that additional data might have. On the other hand, the current presentation demonstrates how the additional data used to develop the new TRF model have extended the expected quality of the TRF model for several additional years over which the old model is clearly not performing nearly as well as during the years used for its development. This proposed rearrangement may or may not be easily achievable, so I would not make it a prerequisite for the acceptance of the article, but it would certainly make it a lot more useful. In general the article is written in very good English, but another read by a separate reader would help correct a few minor issues and make it perfect in that respect. A comment on Fig. 3 is that it is missing a legend, and the caption does not explain what the different colors mean. Adding the values of the displayed slopes would also be nice. Finally, I am chagrined that the authors have left out the required acknowledgements for the SLR and DORIS data that are used to produce the wealth of results that they are presenting! Both communities, ILRS and IDS, depend on getting these acknowledgements and they even have made it simple for the authors to show them exactly how to do it. For example, for the ILRS see:

https://ilrs.cddis.eosdis.nasa.gov/about/cite.html

---

## Author Comment (AC1) · 14 Dec 2018

Dear Reviewer,

thank you very much for your comments that allowed us to improve further the quality of our manuscript. Please find below our responses to your comments and the description of the changes made in the revised manuscript (in blue) following your respective comments (in black). The page (P) and line (L) numbers refer to those given in the discussion paper.

Sincerely yours,

Sergei Rudenko (on behalf of the co-authors)

**Anonymous Referee #1**

Review of "Impact of terrestrial reference frame realizations on altimetry satellites orbit quality, global and regional sea level trends: case ITRF2014 versus ITRF2008" by Sergei Rudenko, Saskia Esselborn, Tilo Schöne, and Denise Dettmering.

This paper discusses what the title says, namely the impact of a new reference system called ITRF2014 on sea level change observed by satellite altimetry. The conclusion is that the errors originating from orbit determination on the observed sea level reduce somewhat. The problem is investigated for TOPEX/Poseidon, and subsequent altimeters Jason-1 and Jason-2. In general, this paper is rather technical and unfortunately not very exiting to read. I have some general comments to improve the quality.

Page 2, lines 1-2: I think it is better to not only mention the uncertainties, but also to mention the physical range of the sea level signal. Another aspect is that all used altimeter satellites only provide coverage between 66N and 66S, a part of the globe is missing, namely in the polar seas and also on the continental shelves.

Certainly, satellite altimeters from the TOPEX/Jason series only provide sea level between 66°S and 66°N. Here we study the impact of the use of different ITRF realizations on the estimation of sea level trends based on altimeter data for these missions, since these are the reference missions for sea level investigations. We have clarified the latitude range by adding the words "(between 66°S and 66°N in latitude)" on P3, L13.

Additionally, we have reformulated the text (P2, L1-3) as follows: "This leads to the changes of interannual global mean sea level of up to 0.06 mm/year for TOPEX/Poseidon, 0.05 mm/year for Jason-1, and up to 0.12 mm/year for Jason-2, i.e. up to 4% of the corresponding sea level signal based on altimetry for time scales of 3 to 8 years. The respective changes of the regional sea level trend on these time scales are up to 0.4 mm/year at the time span from April 1993 to July 2008 and up to 1.0 mm/year at the time span from July 2008 to April 2015."

Page 2, line 12-13: Merge the sentences

Two sentences have been rewritten as "Three new recently released ITRS realizations are ITRF2014 (Altamimi et al., 2016), DTRF2014 (Seitz et al., 2016) and JTRF2014 (Abbondanza et al., 2017)."

Page 2, line 17, "has been shown to have a noticeable impact" noticeable could be read as large or significant, but your study does not show this, the effect of orbit determination uncertainties on sea level rise observed by the altimeters is small. For semantic reasons you could better say, it is observable or detectable, but avoid to suggest that it is significant because it is not.

When speaking about a noticeable impact, we mean the changes of up to ±1.5 mm/yr in the regional sea level rates related to the change from ITRF2000 to ITRF2005 that were found by Beckley et al. (2007). Therefore, we reworded two sentences on P2, L16-19 as follows: "The realization of a terrestrial reference system has been shown (Beckley et al., 2007) to have detectable impact on the regional and global mean sea levels trends derived from altimetry. Thus, they found changes of up to ±1.5 mm/yr in the regional sea level rates related to the change from ITRF2000 to ITRF2005."

Additionally, we have replaced the word "significant" by the words "observable" and "detectable" in the appropriate cases in the manuscript.

Page 2, line 18-19: "While Beckley et al . . . 2014" : In my opinion this sentence does not add new information to your paper. Remove it I would say.

Done.

Page 2, line 31-32: "They found that . . . mm/year" so this is more significant than what you mention, could you perhaps explain why their results differ from yours?

The changes of the global and regional trends strongly depend on the analyzed period of time (please, compare Table 7 and Fig. 7). In fact, for the period 1993 to 2015 we found regional trend changes of up to 0.14 mm/year which are not that different from the 0.2 mm/year given by Zelensky et al. (2018) for the period 1992 to 2016.

To stress that, we have added the following sentence on P13, L16: "The changes of the global and regional trends strongly depend on the analyzed period of time (see Table 7 and Fig. 7)." Additionally, we have added the following text on P13, L19: "The changes of the regional sea level trend (up to 0.14 mm/year) found by us from the analysis of TOPEX, Jason-1 and Jason-2 for the period 1993 to 2015 agree with those (up to 0.2 mm/year) obtained by Zelensky et al. (2018) by using the same altimetry missions for the period 1992 to 2016."

Page 3, line 2-3: "Therefore, in this paper . . . realization", this is repetition of information, these repetitive episodes make the paper uninteresting to read.

We have erased the words "here we analyse the impacts of the change from ITRF2008 to ITRF2014" on P2, L14. Therefore, there is no a repetition any longer at this place.

Page 3, line 7: Do your orbits for T/P, J-1 and J-2 satisfy the criteria of GDR-E which is the latest standard I believe? Where are the differences? Where are the common points? This should be discussed in your paper.

We have added the following text on P4, L22: "Some of the models used by us for POD correspond to Geophysical Data Records (GDR)-E POD standards, some of them, like e.g. ITRF2014, correspond to Precise Orbit Ephemeris (POE)-F standards. At the same time, some of the models used by us, e.g. EOT11a ocean tide model, EIGEN-6S4 Earth gravity field model, GFZ AOD1B RL05 non-tidal atmospheric and oceanic gravity model, differ from those defined in GDR-E and POE-F POD standards details on which can be found at ftp://ftp.ids-doris.org/pub/ids/data/POD_configuration_POEF.pdf."

Page 4, line 1 under table 1: "non-linear station motions" please name it for what it is, it is called a (post) seismic deformation model in ITRF2014, is it the same as what you mention under the third point under table 1?

The non-linear station motions, in case of ITRF2014, are provided by seasonal (annual and semiannual) signals of station positions that were excluded from station positions prior to the determination of station positions and velocities and by post-seismic deformations are provided for stations affected by major earthquakes. Therefore, we have combined three items given on P4, L1-4 in one item.

Page 5, line 2 under table 2: repair this sentence, for instance, the RMS of SLR to T/P did not change significantly when we switched from ITRF2008 to ITRF2014.

We have reformulated this sentence as follows: "We have found that switch from ITRF2008 to ITRF2014 did not change significantly the RMS fits of SLR observations of TOPEX/Poseidon".

Page 5, line 15-16: did you focus on exactly the same ILRS stations and IDS beacons during the comparison? If this is not the case then I wonder about the realism of this comparison.

Yes, we used exactly the same ILRS stations and IDS beacons during the comparison. The increased number of accepted observations, when using ITRF2014, is explained by better fits of observations that were eliminated by a 3-σ criterion, when using ITRF2008 at these arcs. The have added the words "due to better fitting of observations" on P5, L16.

Page 5, line 17-18: It is not entirely clear what you mean with internal consistency of satellite orbits, this should be better explained. During orbit overlaps the trajectory is computed with the same gravity field, the same forcings apply on the satellite, the same station coordinates are used etc, so all possible common effect will cancel.

P5, L17-18: we have replaced this sentence by the following text: "Satellite orbit and adjusted parameters are computed at different arcs using different observations and, in some cases, using different parameterization depending on the amount of available observations. Therefore, though the background models used for orbit computations at orbit overlaps are the same, non-zero differences of satellite coordinates at overlaps are obtained.  We call the differences of satellite coordinates of overlaps *internal* consistency, since the orbits are computed using the same software and the same background models."

In table 3 I recommend to use ITRF 2008 and ITRF2014 consistently where you speak about VER11 and VER13. Below the table you describe exactly, in words, what the readers already see in the table. I do not understand why this information needs to be repeated.

We have replaced VER11 by ITRF2008 and VER13 by ITRF2014 in this table, as suggested by the Reviewer. We have reformulated the last sentence of Table capture as follows: "The percentage of the parameter change by switching from ITRF2008 to ITRF2014 is given in parentheses (positive value indicates an improvement)."

We have excluded the text from P5, L19 to P6, L5, as suggested by the Reviewer, by just leaving the sentence on the along-track overlaps.

Line 14 page 6, what makes a crossover point valid?

We have added the following explanation in the manuscript: "To find valid crossover points, all internal quality criteria, which are part of the GDRs, are checked. In addition, we use only those points which meet the following criteria: deep water (with depth larger than 200 m), wind speed less than 15 m/s, SWH less than 12 m, and crossover difference less than 1.5 m. The latter is especially valid in areas with sea ice occurrence. "

Lines 3-5 page 7: Did you ever investigate the choice of the time variable gravity model behavior on this? The same question could be asked about figure 2.

Yes, we performed similar investigations for studying the impact of geopotential models (Rudenko et al., 2014), and ocean and atmospheric de-aliasing products (Rudenko et al., 2016). In fact, we mentioned that on P9, L1-2. Now we slightly reformulated this text and moved it to P3, L13.

Figure 3 on page 9: this is too much of an excel like image in my opinion, the quality should be improved.

We have improved the quality of Fig. 3 by adding a legend, the values of the displayed slopes, and the description of the X axis.

Table 4 and 5: either use ver11 and ver13, or use itrf2008 and itrf2014, but not the double referencing to ver11/itrf2008 and ver13/itrf2014 like you do now.

We have excluded VER11 and VER13 from Tables 4-6, and kept just ITRF realization names, as suggested by the Reviewer.

Line 2-3 below table 5: repetition of information, see earlier remarks.

We would prefer to keep the discussion of this table.

Line 3 below table 5: TOPEX is the NASA Altimeter and TOPEX/Poseidon is the mission. I recommend that you use abbreviations like (T/P, J1 and J2). Please change all TOPEX references in the manuscript when you are referring to the mission. Only use TOPEX when you refer to the NASA altimeter on T/P.

We have used in our analysis the data only of TOPEX altimeter for the TOPEX/Poseidon satellite. Therefore, we have replaced the words "TOPEX/Poseidon" by "TOPEX" in all cases, when it concerns an

altimeter. Additionally, we have replaced the word "satellite" by the word "mission" in Tables 4-6. For clarity, we would prefer to use the complete names of altimeter satellites instead of their abbreviations suggested by the Reviewer. Additionally, it might make it difficult to find information related to a specific mission, if abbreviations are given.

Line 10 page 10: The concept of GCEs refers back to Rosborough 1987 who should be referenced. He was the first to start with geographically correlated orbit errors.

We agree with the Reviewer. Therefore, we have added the words "Following the theory of Rosborough (1986)," before the words "The MMXO provides" and the following reference:

Rosborough, G.W.: Satellite orbit perturbations due to the geopotential, University of Texas, Center for Space Research Report CSR-86-1, January, 1986.

Page 11, line 14, use the \times symbol in LaTeX when you write 1x1

We have changed this, as suggested on this line and line 15 on the same page.

Line 9 and 10 on Page 13: TOPEX I and TOPEX II should in my opinion be referred to as TOPEX altimeter side A and side B.

The periods have been defined from the tendency of the mean height differences between the radial orbit components derived from ITRF2008 versus ITRF2014 orbits (Figure 7). They are not connected to the changes on the altimeter platform, but rather to the changes in the observation network. The switch between TOPEX side A and side B took place in February of 1999. To clarify this fact we have changed the sentence on P13, L7-8 to: "In order to study these effects in detail we define the following four periods which are covered by the denoted missions from the inflection points of the time series".

Section 7, conclusions. I do not get the feeling that sea level rise estimates significantly change when ITRF2014 is introduced. The GMSL value is still 3.1+/-0.4 mm/yr (see the university of Colorado website) regardless whether we use the ITRF2008 or the ITRF2014 orbits. What you did improve are the DORIS and SLR residuals, but the improvements are small. Figure 8 shows that there are hemispherical like offsets developing over time, you may what to treat this as a geocenter type of discussion. This will affect the regional sea level estimates, which have larger ranges and error bars.

We have reformulated the text on P17, L3-14 as follows: "The change from ITRF2008 to ITRF2014 orbits has only minor effects on the estimation of regional and global sea level trends over the 22-year time series from 1993 to 2015. However, on interannual time scales (3-8 years) large scale coherent trend patterns are observed that seem to be connected to drifts between the origins of the tracking stations networks. This leads to changes of the global interannual trends of up to 0.06 mm/year for TOPEX, 0.05 mm/year for Jason-1, and up to 0.12 mm/year for Jason-2 which corresponds to changes of up to 4% of the actual sea level trends from altimetry. The respective changes of the regional sea level trend reach maximum values of 0.4 mm/year for TOPEX, of 0.5 mm/year for Jason-1 and of 1.0 mm/year for Jason-2. While till 2008 regional sea level drifts are mainly related to uncertainties of the z-coordinate of the origin, recently uncertainties of the horizontal coordinates have become increasingly important. This shows the effects of the increasing uncertainties of the tracking station positions and velocities in

ITRF2008 after the year 2008 on the estimated regional sea level trends. For this period the user requirements for the error of the regional mean sea level (Ablain et al., 2015) cannot be met everywhere when using ITRF2008 orbits."

"Geographically" is hyphenated over a page break (16 and 17).

This has been fixed.

**Additionally we have made the following changes:**

P1: we have changed the words "case ITRF2014 versus ITRF2008" to the words "a switch from ITRF2008 to ITRF2014" in the title of the manuscript.

A missing serial (Oxford) comma has been added in all necessary cases.

We have replaced "mm/y" and "mm/year" by "mm/yr" for consistency.

P1, L10: we have replaced the words "improves orbit quality" by the words "improves the orbit quality".

P2, L31: we have corrected the typo by replacing "over 1993-2015" by "over 1992-2016".

P2, L31: we have replaced the words "causes the impact on" by the word "impacts".

P2, L32: we have corrected the typo by replacing 0.3 mm/year by 0.2 mm/year.

P3, L1: we have replaced "ITRF" by "ITRS".

P3, L22: we words "is discussed in Sect." have been corrected by the words "are discussed in Sect.".

P4, L16: we have added the word "the" before the words ""Earth Parameter and Orbit System – Orbit Computation (EPOS-OC)" software".

P4, L21-22: the words "(version 13)" have been added after the words "VER13" to explain the abbreviation.

P4, L23: we have replaced the word "indication" by the word "indicator".

P6, L7: the word "the" has been added before the words "ESA CCI Sea Level v2 ECV data".

P6, L14: we have replaced the words "In average" by the words "On average".

P9, capture of Figure 3: the words "all three missions" have been replaced by the words "the TOPEX (TP), Jason-1 (J1) and Jason-2 (J2) missions".

P11, L10: We have reformulated this sentence as follows: "The main focus of this analysis is on time scales of more than one year."

P13, capture of Table 7: to make it more clear, we have reformulated the sentence "The percentage of the given value relative to the values derived from altimetry data is given in brackets for comparison." As

follows: "The percentage of the ITRF-related changes relative to the total signal measured by altimetry is given in brackets for comparison."

P13, L6: we have replaced the words "use of the ITRF2014 realization" by the words "switch from the ITRF2008 to the ITRF2014 realization".

P14, L1-9: we have reformulated the text as follows: "The uncertainties of global mean sea level trends due to the TRF realization have decreased considerably during the last decades. Beckley et al. (2007) reported global mean sea level changes of 0.44 mm/year related to the change from ITRF2000 to ITRF2005. The change from ITRF2005 to ITRF2008 still lead to apparent global sea level drifts of 0.05 mm/year (Couhert et al., 2015) at the decadal time scale. For the change from ITRF2008 to 2014, Zelensky et al. (2018) - based on five GSFC orbits - estimated an uncertainty of the global mean sea level of 0.03 mm/year.  According to our studies the global mean sea level trend for the 22 years series is hardly impacted at all. This is a major improvement with respect to the previous TRF realizations. The uncertainties of the global mean sea level trends have been dominated by uncertainties of the z-coordinate of the origin (Beckley et al., 2007). From our analyses we observe corresponding patterns till 2008, even though the amplitudes decreased by factor four.  After 2008 the uncertainties of the horizontal coordinates of the origin increasingly impact regional sea level drifts."

P14, L3: the word "lead" has been corrected by the word "leads".

P16, L11: the word "shows" has been corrected by the word "show".

P16, L14: the words "usage of ITRF2014" have been replace by the words "switch from ITRF2008 to ITRF2014".

P17, L17: we have added the following sentence: "To minimize errors caused by the extrapolation of station velocities beyond the time span at which they had been derived, ITRS realizations should be regularly (at least every 5-6 years) updated."

P17, L21: we have replaced the word "study" by the word "research".

P17, L22: we have added the following sentence: "All authors read and approved the final manuscript."

P17, L29: we have added the following text: "The authors are grateful to K.-H. Neumayer and J.-C. Raimondo for preparing some input data used in this study. The authors thank the editor and two referees for their comments that allowed them to improve this paper."

P19, L11 and L14: we have replaced the full journal name "IEEE Transactions on Geoscience and Remote Sensing" by its abbreviation "IEEE T. Geosci. Remote".

P19, L16: we have added the following reference for GFZ VER13 orbits: Rudenko, S., Schöne, T., Esselborn, S., and Neumayer, K.: GFZ VER13 SLCCI precise orbits of altimetry satellites ERS-1, ERS-2, Envisat, TOPEX/Poseidon, Jason-1, and Jason-2 in the ITRF2014 reference frame, GFZ Data Services, http://doi.org/10.5880/GFZ.1.2.2018.003, 2018.

P19, L27: we have replaced the full journal name "ADVANCES IN SPACE RESEARCH" by its abbreviation "Adv. Space Res.".

---

## Author Comment (AC2) · 14 Dec 2018

**epavlis@umbc.edu**

Dear Reviewer,

thank you very much for your comments that allowed us to improve further the quality of our manuscript. Please find below our responses to your comments and the description of the changes made in the revised manuscript (in blue) following your respective comments (in black). The page (P) and line (L) numbers refer to those given in the discussion paper.

Sincerely yours,

Sergei Rudenko (on behalf of the co-authors)

This article is presenting an important piece of work that even though it is not groundbreaking research, it is of great value to the users of altimetry data and products. When one reads this article one has the feeling that the intent of the authors is to record in high detail the work that was done to compare the difference in the "products" of the authors' lab between two reductions of the same (or nearly so) data with two different reference frame models. The analysis though is over different periods of time, something quite natural, since you only change reference frame models once the prior version becomes obsolete. However, the presentation herein would have benefited from a slightly modified presentation of the differences: the one that is already in this version of the article, and a parallel one where the metrics are formed, presented and evaluated over exactly the same time period. This arrangement would remove the effect of the additional data used to develop the products beyond the validity period of the old TRF, along with any issues that that additional data might have. On the other hand, the current presentation demonstrates how the additional data used to develop the new TRF model have extended the expected quality of the TRF model for several additional years over which the old model is clearly not performing nearly as well as during the years used for its development. This proposed rearrangement may or may not be easily achievable, so I would not make it a prerequisite for the acceptance of the article, but it would certainly make it a lot more useful.

This is an interesting idea. In fact, the analyses and statistics for the TOPEX/Poseidon mission show the differences between the two ITRF realizations till 2005, which is still in the period from which the data for the generation of ITRF2008 were used. On the contrary, the results for the Jason-2 mission (since 2009.0) are outside this period. The temporal behavior of the parameters before 2009.0 and after this time instant is good visible in Fig. 1, 3, 4, 6, and 7.

To stress that, we have replaced the text on P17, L15-16 with the following text: "Our analyses show that the use of ITRF2014 instead of ITRF2008 slightly improves the satellite orbits as well as the derived sea

level values since 1993. The analyses and statistics for TOPEX/Poseidon show the differences between the two ITRF realizations till 2005. More evident improvements are found from 2009.0 for Jason-1 and, in particular, for Jason-2."

In general the article is written in very good English, but another read by a separate reader would help correct a few minor issues and make it perfect in that respect.

We have fixed a few errors in English and improved English in the manuscript.

A comment on Fig. 3 is that it is missing a legend, and the caption does not explain what the different colors mean. Adding the values of the displayed slopes would also be nice.

We have added a legend, the values of the displayed slopes, as well as the description of the X axis in Fig. 3. We hope, the quality of the figure improved.

Finally, I am chagrined that the authors have left out the required acknowledgements for the SLR and DORIS data that are used to produce the wealth of results that they are presenting! Both communities, ILRS and IDS, depend on getting these acknowledgements and they even have made it simple for the authors to show them exactly how to do it. For example, for the ILRS see: https://ilrs.cddis.eosdis.nasa.gov/about/cite.html

We have added the following citations for ILRS and IDS, respectively:

Pearlman, M.R., Degnan, J.J., and Bosworth, J.M.: The International Laser Ranging Service, Adv. Space Res., 30 (2), 135--143, doiI:10.1016/S0273-1177(02)00277-6, 2002.

Willis, P., Fagard, H., Ferrage, P., Lemoine, F.G., Noll, C.E., Noomen, R., Otten, M., Ries, J.C., Rothacher, M., Soudarin, L., Tavernier, G., Valette, J.J.: The International DORIS Service (IDS): Toward maturity, in DORIS: Scientific Applications in Geodesy and Geodynamics, P. Willis (Ed.), Adv. Space Res., 45(12),1408-1420, doi: 10.1016/j.asr.2009.11.018, 2010.

Additionally we made the changes described at the end of our rebuttal letter to the interactive comment of Reviewer 1.